# Improving the Utilization of STRmix™ Variance Parameters as Semi-Quantitative Profile Modeling Metrics

**DOI:** 10.3390/genes14010102

**Published:** 2022-12-29

**Authors:** Kyle Duke, Steven Myers, Daniela Cuenca, Jeanette Wallin

**Affiliations:** California Department of Justice, Richmond, CA 94804, USA

**Keywords:** STRmix, variance, diagnostic, profile modeling, probabilistic genotyping

## Abstract

Distributions of the variance parameter values developed during the validation process. Comparisons of these prior distributions to the run-specific average are one measure used by analysts to assess the reliability of a STRmix deconvolution. This study examined the behavior of three different STRmix variance parameters under standard amplification and interpretation conditions, as well as under a variety of challenging conditions, with the goal of making comparisons to the prior distributions more practical and meaningful. Using information found in STRmix v2.8 Interpretation Reports, we plotted the log_10_ of each variance parameter against the log_10_ of the template amount of the highest-level contributor (Tc) for a large set of mixture data amplified under standard conditions. We observed nonlinear trends in these plots, which we regressed to fourth-order polynomials, and used the regression data to establish typical ranges for the variance parameters over the Tc range. We then compared the typical variance parameter ranges to log_10_(variance parameter) v log_10_(Tc) plots for mixtures amplified and interpreted under a variety of challenging conditions. We observed several distinct patterns to variance parameter shifts in the challenged data interpretations in comparison to the unchallenged data interpretations, as well as distinct shifts in the unchallenged variance parameters away from their prior gamma distribution modes over specific ranges of Tc. These findings suggest that employing empirically determined working ranges for variance parameters may be an improved means of detecting whether aberrations in the interpretation were meaningful enough to trigger greater scrutiny of the electropherogram and genotype interpretation.

## 1. Introduction

The forensic genetics community currently relies on PCR-amplified short tandem repeat (STR) DNA profiles to assess the source of biological evidence associated with criminal activity. Such biological material is often presented as a mixture of DNA from different individuals, necessitating deconvolution of the STR electropherogram results [1]. An integral part of forensic DNA mixture deconvolution is modeling STR peak amplitude, which varies with DNA input [2,3]. Binary DNA interpretation procedures often address this peak height variation through the application of threshold-based heuristics that establish lower bounds for per-contributor intra-locus peak height balance [4,5]. In contrast, continuous probabilistic genotyping approaches to interpretation rely on probability distributions to represent variance expectations [6,7].

In the case of STRmix™ Probabilistic Genotyping Software, which applies a continuous profile model to forensic DNA mixture data, peak height variation is integrated into STR profile analysis through the application of dynamic variance parameters that are inversely proportional to peak height [8,9]. Higher peak heights are accompanied by smaller variance parameters, leading to less allowance for differences between observed peak heights and those modeled by the software. More specifically, the spread of the lognormal distribution of [observed RFU height]/[modeled RFU height] increases with an increase in a variance parameter and decreases with an increase in peak height, as represented mathematically below:(1)$$\log_{10}\left(\mathrm{Observed}\;\mathrm{RFU}/\mathrm{Modeled}\;\mathrm{RFU}\right)\;\sim \;N\left(0,\;\left[\mathrm{Variance}\;\mathrm{parameter}\right]/\left[\mathrm{Relevant}\;\mathrm{peak}\;\mathrm{height}\right]\right)$$

If the peak being modeled is an allele, the relevant peak height is always the height of the allele itself modeled by STRmix v2.8 (i.e., the expected height). Alternatively, if the modeled peak is stutter, the relevant peak height is set by the software user and can be either the expected height of the stutter peak itself or the observed height of the allele giving rise to the stutter. Given a hypothesized set of genotypes, a particular peak may also be composed of signal from both allele and stutter peaks, and in these cases, a shifted lognormal model that combines the relevant allele and stutter lognormals together is utilized [9].

In the preliminary stages of STRmix v2.8 implementation, a large training set of single-source profiles with known types is utilized to construct prior gamma distributions of the variance parameters (see Figure 1). These prior gamma distributions, created by the Model Maker module in STRmix, are reproduced on every STRmix Interpretation Report, along with the average variance parameters for the interpretation. The creators of STRmix define the genotype probability distributions (GPDs), mixture proportions, and per-locus likelihood ratios (LRs) as primary diagnostics of an interpretation, whereas the average variance parameters are among the secondary STRmix diagnostics that have less well-defined acceptable ranges but provide information about profile modeling efficacy [10]. Comparison of the average variance parameter for a particular interpretation to the mode of the prior gamma distribution provides a point of reference as to whether greater allowance for variation was required to model the data than was needed for the training set.

However, the comparison of average variance parameters to the prior probability distributions appears to have the same need for threshold calculations that is so central to binary interpretation paradigms. Namely, in order to utilize the average variance parameter values as semi-quantitative diagnostics, it would be important to estimate at what values the average allele and/or stutter variance parameters are far enough away from their prior distribution modes that the STRmix output deserves closer inspection.

Here, we present a systematic examination of STRmix v2.8 allele, reverse stutter (−1 STR repeat), and forward stutter (+1 STR repeat) variance parameters. The goals were to characterize any apparent trends in the variance parameters across a variety of single-source and mixture data, as well as to develop a typical range of parameter values by comparison under standard (i.e., pristine template DNA) and challenging amplification and interpretation conditions. It should be noted that, while both the lower and upper range of variance parameter values are relevant metrics of system performance, STRmix v2.8 has a default lower bound for variance parameters of (0.5) × (prior distribution mode), so we focused exclusively on the upper range in this work.

The value of this study is to increase the usefulness of MCMC (Markov Chain Monte Carlo) summary diagnostics for the forensic DNA community utilizing probabilistic genotyping software. Knowing more about how STRmix variance parameters behave across a large mixture dataset drives empirically supported, reliable assessments of data for forensic case interpretation, addressing related concerns on factor space and likelihood ratio “trustworthiness” as outlined in the recent NISTIR 8351-DRAFT [11].

## 2. Materials and Methods

### 2.1. Construction, Amplification, Capillary Electrophoresis, and Analysis of Unchallenged DNA Samples

Buccal cell DNA was collected with informed consent from healthy, unrelated laboratory volunteers and extracted using the PrepFiler™ DNA Extraction Kit (Life Technologies, South San Francisco, CA, USA). Samples were quantified for extraction yield using Applied Biosystems™ Quantifiler™ Trio DNA Quantification kit, amplified using Applied Biosystems™ GlobalFiler™ PCR Amplification kit with the manufacturer’s 28-cycle thermal cycling protocol and capillary electrophoresed (CE) on an Applied Biosystems™ 3500 Genetic Analyzer (1.2 kV injection for 15 s) using POP-6™ Polymer (Life Technologies). Raw data from all amplified samples were subsequently analyzed in GeneMapper ID-X v1.6 Software (Life Technologies) at channel-specific analytical thresholds of 51 RFU (blue), 71 RFU (green), 35 RFU (yellow), 41 RFU (red), and 61 RFU (purple) and analyzed with STRmix v2.8 Probabilistic Genotyping Software (NicheVision, Akron, OH, USA) using laboratory-validated settings, which included stutter models for −1/+1, −2/+2, −0.5/+0.5, and −1.5 STR repeats. The −1 repeat stutter variance was set to be inversely proportional to the observed height of the parent allele; all other stutter variances were set to be inversely proportional to the expected stutter peak height (personal communication with STRmix support staff, 1 September 2020). MCMC accepts per each of the 8 chains were set to 10,000 burn-in/50,000 post burn-in for single-source, 2-person mixtures and 3-person mixtures, and to 200,000 burn-in/1,000,000 post burn-in for 4-person mixtures [12]. The Gelman-Rubin autocontinue option was activated for all runs, and was set to add 10,000 post burn-in accepts to any MCMC initially producing a Gelman-Rubin convergence diagnostic in excess of 1.2. See Appendix A for the relevant STRmix interpretation data.

The unchallenged data consisted of single-source samples and 2-, 3-, and 4-person mixtures. Single-source samples (Table 1) included 26 amplifications from 16 individuals, ranging from 8 ng to 63 pg input template DNA for PCR. Unchallenged mixtures at a variety of DNA input amounts and mixture ratios (Table 2) were constructed by combining the indicated amounts of template. See Appendix A for anonymized contributor genotypes.

### 2.2. Construction, Amplification, CE, and Analysis of Challenged DNA Samples

Data from mixtures challenged by inhibition, degradation, signal saturation, and underestimated number of contributors (NOC) were included for comparison to the unchallenged data. These challenged samples were processed as described in Section 2.1, with any exceptions noted below.

Inhibited mixtures were constructed by combining the mixture components as indicated in Table 3 and spiking the subsequent 28-cycle GlobalFiler amplification with the amount of inhibitor (either hematin or humic acid) necessary to achieve the concentration indicated in the table.

To prepare degraded mixtures, degraded single-source DNA was first produced by subjecting buccal samples on BODE buccal collectors to a dry heat bath set to 90 °C for the amounts of time indicated in Table 4. After extraction and quantification of the degraded single-source mixture components, the extracts for each component were amplified, electrophoresed, and paired to each other such that the paired components were degraded to a similar extent, as indicated by an exponential fitting of peak heights. The paired components were then combined in the ratios indicated in Table 5. Note that the inhibited and degraded data sets have the same mixture donors, although this set of donors is different from the donor sets used for the unchallenged data set.

Signal-saturated mixtures were produced by combining the extracts from the unchallenged dataset in the ratios and amounts indicated in Table 6.

A subset of the unchallenged mixtures in Table 2 were re-analyzed with STRmix set to one or two contributors fewer than the true number, as indicated in Table 7. These mixtures were qualitatively categorized by an experienced DNA analyst as ambiguous in contributor number due to intralocus peak imbalances and/or sub-threshold peak information.

An additional set of challenged mixtures was constructed by combining commercial preparations of cell line DNA in the proportions indicated in Table 8. The cell line DNA preparations used to generate this data were HL60 (NIST, Gaithersburg, MD, USA), CEPH 1347-02 (Thermo Fisher Scientific, Waltham, MA, USA), and 2800 M (Verogen, San Diego, CA, USA). This set of mixtures was intended for inclusion in the unchallenged data set, but unintuitive results were observed upon interpretation of the mixtures that indicated they did not fit the STRmix profile model.
genes-14-00102-t001_Table 1Table 1Unchallenged single-source samples. Upon examination of the CE data, a signal-saturated peak was observed in one of the 8 ng replicate amplifications; for purposes of plotting, data for this amplification was grouped with the saturated mixtures in Table 6.StudyNumber of SamplesInput AmountsReplicate AmplificationsReplicate STRmix InterpretationsSingle-source, nominal-input142 ng110Single-source dilution series (higher level)28 ng, 4 ng, 2 ng, 1 ng, 500 pg, 250 pg21Single-source dilution series (lower level)2125 pg, 63 pg41
genes-14-00102-t002_Table 2Table 2Composition of unchallenged mixtures. The input amounts listed are for total DNA. Note that, to include STRmix run-to-run variation in the dataset, ten replicate interpretations were performed for each mixture, at each template level.Donor Number (Donor Set)Mixture RatioInput AmountsReplicate Amplifications2-person (set 1)9:12 ng, 1 ng, 870 pg, 750 pg, 500 pg, 380 pg, 250 pg, 125 pg, 63 pg22-person (set 1)49:12.5 ng, 1.9 ng, 1.25 ng, 625 pg, 313 pg22-person (set 1)99:12.5 ng, 1.25 ng, 625 pg22-person (set 2)1:1800 pg, 400 pg, 200 pg, 100 pg, 50 pg, 25 pg12-person (set 2)3:1800 pg, 400 pg, 348 pg, 300 pg, 200 pg, 152 pg, 100 pg, 50 pg, 25 pg13-person (set 1)3:2:11.2 ng, 600 pg, 522 pg, 450 pg, 300 pg, 228 pg, 150 pg, 75 pg, 38 pg23-person (set 1)10:5:13.2 ng, 1.6 ng, 1.4 ng, 1.2 ng, 800 pg, 608 pg, 400 pg, 200 pg, 100 pg23-person (set 1)100:100:41.28 ng, 625 pg, 325 pg23-person (set 2)1:1:11.2 ng, 600 pg, 300 pg, 150 pg, 75 pg, 38 pg13-person (set 2)3:2:11.2 ng, 522 pg, 300 pg, 150 pg, 38 pg23-person (set 2)10:5:13.2 ng, 1.4 ng, 800 pg, 400 pg, 100 pg23-person (set 2)100:100:41.28 ng, 638 pg, 319 pg24-person (set 1)4:3:2:12 ng, 1 ng, 870 pg, 750 pg, 500 pg, 380 pg, 250 pg, 125 pg, 63 pg24-person (set 1)10:5:2:13.6 ng, 1.8 ng, 1.6 ng, 1.4 ng, 900 pg, 684 pg, 450 pg, 225 pg, 113 pg24-person (set 1)100:100:100:61.28 ng, 625 pg, 325 pg24-person (set 2)1:1:1:11.6 ng, 800 pg, 400 pg, 200 pg, 100 pg, 50 pg14-person (set 2)4:3:2:12 ng, 870 pg, 500 pg, 250 pg, 63 pg24-person (set 2)10:5:2:13.6 ng, 1.6 ng, 900 pg, 450 pg, 113 pg24-person (set 2)100:100:100:61.28 ng, 638 pg, 319 pg2
genes-14-00102-t003_Table 3Table 3Sample composition of inhibited mixtures. All inhibited mixtures were amplified and interpreted once at a total DNA input amount of 3 ng.Donor NumberMixture RatioTreatment2-person3:1Hematin: 400 µM, 475 µM, 550 µM, 625 µM, 700 µM10:1Hematin: 400 µM, 475 µM, 550 µM, 625 µM, 700 µM3:1Humic acid: 200 ng/µL, 250 ng/µL, 300 ng/µL, 350 ng/µL, 400 ng/µL10:1Humic acid: 200 ng/µL, 250 ng/µL, 300 ng/µL, 350 ng/µL, 400 ng/µL3-person3:2:1Hematin: 400 µM, 475 µM, 550 µM, 625 µM, 700 µM10:5:1Hematin: 400 µM, 475 µM, 550 µM, 625 µM, 700 µM3:2:1Humic acid: 200 ng/µL, 250 ng/µL, 300 ng/µL, 350 ng/µL, 400 ng/µL10:5:1Humic acid: 200 ng/µL, 250 ng/µL, 300 ng/µL, 350 ng/µL, 400 ng/µL4-person4:3:2:1Hematin: 400 µM, 475 µM, 550 µM, 625 µM, 700 µM10:5:2:1Hematin: 400 µM, 475 µM, 550 µM, 625 µM, 700 µM4:3:2:1Humic acid: 200 ng/µL, 250 ng/µL, 300 ng/µL, 350 ng/µL, 400 ng/µL10:5:2:1Humic acid: 200 ng/µL, 250 ng/µL, 300 ng/µL, 350 ng/µL, 400 ng/µL
genes-14-00102-t004_Table 4Table 4Dry heat exposure times for degraded mixture components.Dry Heat Treatment NumberDry Heat Exposure Time15.75 h212.13 h319.42 h427.73 h537.32 h648.50 h761.70 h877.52 h996.83 h10120.93 h11151.85 h12192.97 h13250.32 h14335.88 h
genes-14-00102-t005_Table 5Table 5Sample composition of degraded samples. Dry heat treatment numbers refer to those in Table 4 and, for mixtures, are listed in order according to which treated components were paired together. All degraded single-source samples were amplified at a DNA input amount of 2 ng, and all degraded mixtures were amplified at a total DNA input of 8 ng.Donor NumberMixture RatioC1 Dry Heat TreatmentsC2 Dry Heat TreatmentsC3 Dry Heat TreatmentsC4 Dry Heat TreatmentsSingle source #1-1,3,4,5,6,7,9,13,10---Single source #2-1,2,3,4,5,6,8,10,14---Single source #3-1,2,4,5,6,9,10,11,12---Single source #4-1,2,4,5,8,9,11,12,13---2-person3:11,3,4,5,6,7,9,13,101,2,3,4,5,6,8,10,14--10:11,3,4,5,6,7,9,13,101,2,3,4,5,6,8,10,14--3-person3:2:11,3,4,5,6,7,9,13,101,2,3,4,5,6,8,10,141,2,4,5,6,9,10,11,12-10:5:11,3,4,5,6,7,9,13,101,2,3,4,5,6,8,10,141,2,4,5,6,9,10,11,12-4-person4:3:2:11,3,4,5,6,7,9,13,101,2,3,4,5,6,8,10,141,2,4,5,6,9,10,11,121,2,4,5,8,9,11,12,1310:5:2:11,3,4,5,6,7,9,13,101,2,3,4,5,6,8,10,141,2,4,5,6,9,10,11,121,2,4,5,8,9,11,12,13
genes-14-00102-t006_Table 6Table 6Sample composition of signal-saturated mixtures. Note that the sets of donors used for these amplifications are the same as the donor sets in Table 1.Donor Number (Donor Set)Mixture RatioInput Amounts (Total DNA)2-person (set 1)9:128 ng2-person (set 1)99:125.5 ng2-person (set 2)1:129.3 ng2-person (set 2)3:120.9 ng3-person (set 1)3:2:124 ng3-person (set 1)10:5:129 ng3-person (set 1)100:100:428.5 ng3-person (set 2)1:1:120.4 ng3-person (set 2)3:2:117.8 ng3-person (set 2)10:5:112.7 ng3-person (set 2)100:100:417.1 ng4-person (set 1)4:3:2:132.5 ng4-person (set 1)10:5:2:129.4 ng4-person (set 1)100:100:100:630 ng4-person (set 2)1:1:1:120.3 ng4-person (set 2)4:3:2:115.4 ng4-person (set 2)10:5:2:111.9 ng4-person (set 2)100:100:100:622.0 ng
genes-14-00102-t007_Table 7Table 7Sample composition of mixtures analyzed with STRmix set to one or two contributors less than the ground truth number, as indicated.Ground Truth Donor Number (Donor Set)STRmix Donor Number Setting(NOC-1 or NOC-2)Mixture RatioInput Amounts (Total DNA)2-person (set 1)149:1313 pg2-person (set 1)199:12.5 ng, 1.25 ng3-person (set 1)23:2:11.2 ng, 600 pg, 522 pg, 450 pg, 300 pg, 228 pg, 150 pg, 75 pg, 38 pg3-person (set 2)23:2:138 pg3-person (set 1)210:5:1800 pg, 608 pg, 400 pg, 200 pg, 100 pg3-person (set 2)210:5:1100 pg3-person (set 1)2100:100:41.28 ng, 625 pg, 325 pg3-person (set 2)2100:100:41.28 ng, 625 pg, 325 pg4-person (set 1)24:3:2:1125 pg, 63 pg4-person (set 1)210:5:2:1113 pg4-person (set 2)210:5:2:1113 pg4-person (set 1)34:3:2:12 ng, 1 ng, 870 pg, 750 pg, 500 pg, 380 pg, 250 pg4-person (set 2)34:3:2:163 pg4-person (set 1)310:5:2:13.6 ng, 1.8 ng, 1.6 ng, 1.4 ng, 684 pg, 450 pg, 225 pg4-person (set 2)310:5:2:1450 pg4-person (set 1)3100:100:100:61.28 ng, 638 ng, 319 ng4-person (set 2)3100:100:100:61.28 ng, 625 ng, 325 ng
genes-14-00102-t008_Table 8Table 8Sample composition of cell line DNA mixtures. The cell lines listed in the first column correspond to the mixture ratios listed in column 2.Donor NumberMixture RatioInput AmountsReplicate Amps2-person(CEPH 1347-02, HL60)9:12 ng, 1 ng, 870 pg, 750 pg, 500 pg, 380 pg, 250 pg, 125 pg, 63 pg249:12.5 ng, 1.9 ng, 1.25 ng, 625 pg, 313 pg299:12.5 ng, 1.25 ng, 625 pg21:1800 pg, 400 pg, 200 pg, 100 pg, 50 pg, 25 pg13:1800 pg, 400 pg, 348 pg, 300 pg, 200 pg, 152 pg, 100 pg, 50 pg, 25 pg13-person(2800 M, HL60, CEPH 1347-02)3:2:11.2 ng, 600 pg, 522 pg, 450 pg, 300 pg, 228 pg, 150 pg, 75 pg, 38 pg210:5:13.2 ng, 1.6 ng, 1.4 ng, 1.2 ng, 800 pg, 608 pg, 400 pg, 200 pg, 100 pg2100:100:41.28 ng, 625 pg, 325 pg21:1:11.2 ng, 600 pg, 300 pg, 150 pg, 75 pg, 38 pg13:2:11.2 ng, 522 pg, 300 pg, 150 pg, 38 pg210:5:13.2 ng, 1.4 ng, 800 pg, 400 pg, 100 pg2100:100:41.28 ng, 638 pg, 319 pg2

## 3. Results

### 3.1. Trends in Allele and Stutter Variances with Increasing Peak Height

The inverse proportionality between the lognormal variances in Equation (1) and peak heights (in RFU) would suggest that plots of the average variance parameters against the average “Template (rfu)” values, both of which are found in the STRmix Interpretation Report, hold promise in understanding variance behavior. Given the ranges of the variance and template levels that may be observed in profiles, we found it easier to visualize trends between these variables by plotting their base-10 logarithms against each other. Additionally, in order to avoid skewing plots in favor of DNA mixtures with more contributors, we plotted only the “Template (rfu)” value for the highest-level contributor (which we have termed Tc).

Figure 2a–c show the results of plotting log_10_(variance parameter) against log_10_(Tc) for the unchallenged data set in relation to allele variance, reverse stutter variance, and forward stutter variance, respectively. To varying degrees, the data trends across the full Tc range are not linear for all three variances. Using the LINEST function of Excel, we found that a fourth-order polynomial regression provided the best visual fit to each dataset. Quantitative assessments of how well the regression predicts the log_10_(variance) can be found in Appendix A, Table 9 and Appendix A. No significant deviations from normality around the regression lines were observed in the residuals of the log_10_(variance) data when evaluated using the Jarque-Bera test [13] (see Table 9; all *p* > 0.01). Appendix A displays the 99% 2-sided confidence intervals around the regression lines. Appendix A includes the coefficients of determination (R^2^) and F statistic *p*-values for the overall regressions, as well as the 99% 2-sided confidence intervals and T statistic *p*-values for the individual polynomial regression coefficients (*β_i_*_=0*to*4_). All R^2^ values are below 0.5, but the F statistics suggest they are all significantly different from 0 (*p* < 0.01). The individual coefficients for the allele and reverse stutter regressions are also significantly different from 0 (*p* < 0.01; 99% confidence intervals never included 0). For the forward stutter, only coefficient *β*_4_ is significantly different from 0. This is consistent with a visual assessment of the forward stutter data, where a clear bias away from the gamma mode is observed, but Tc appears to have no effect on the variance for log_10_(Tc) < 3.

The 99th percentiles were calculated based upon the fitted polynomial regressions + 2.326 standard deviations (SD) of the residuals (Table 9). In Figure 2a–c, the regression and 99th percentile lines are plotted in red. In addition, a second set of horizontal black lines was plotted to show bands of expected variation that are based on the shape of the prior gamma distributions for the modeled allele and stutter types (see Table 10 for prior gamma distribution information). Specifically, the solid black lines are the modes of the prior gamma distributions. A similar calculation was performed to produce the dotted black lines, given a static variance parameter equal to the 99th percentile of the corresponding prior gamma distribution.

Comparing the black and red bands of expected variation in Figure 2 provides information about whether the variances from completed interpretations tend to mirror the prior gamma distributions of the variance parameters. While the allele variance regression in Figure 2 had a very similar trend to the prior gamma distribution mode line, the reverse and forward stutter variance parameters were very often elevated above their prior gamma modes. It is worth noting, however, that 100 percent of the unchallenged data fell below the lines defined by the 99th percentile of each prior gamma distribution.
genes-14-00102-t009_Table 9Table 9A summary of fourth-order polynomial regression information for the allele, reverse stutter, and forward stutter variance plots from Figure 2 (corresponding to red lines).
AlleleReverse StutterForward StutterPolynomial regression formulay = 0.1122x^4^ − 1.2206x^3^ + 4.8535x^2^ − 8.2618x + 5.5312y = −0.2892x^4^ + 3.3258x^3^ − 13.619x^2^ + 23.456x − 13.404y = −0.0348x^4^ + 0.313x^3^ − 1.0891x^2^ + 1.7016x − 0.1678Jarque-Bera test for normality of the residuals*p* = 0.1503*p* = 0.2395*p* = 0.027599th Percentile(+2.326 SD)+0.2156+0.2755+0.1779
genes-14-00102-t010_Table 10Table 10A summary of prior gamma distribution information for the allele, reverse stutter, and forward stutter variance parameters used to generate the data plotted in Figure 2 (corresponding to black lines).
AlleleReverse StutterForward Stutterα3.8911.5571.526β1.1316.4364.552Mode3.2703.5852.39499th Percentile11.1637.2426.06

### 3.2. Trends in Allele and Stutter Variance under Challenging Amplification/Interpretation Conditions

The bands of expected variation in the unchallenged plots from Figure 2 were used as a benchmark from which to assess the effects of challenging amplification and interpretation conditions on allele and stutter variance parameters. Overlaying the bands from Figure 2 onto similar plots of log_10_(variance parameter) v log_10_(Tc) for the challenged datasets allowed for a direct visualization of variance shifts.

Figure 3 is an overlay of the unchallenged bands of expected variation from Figure 2 onto log_10_(variance parameter) v log_10_(Tc) plots for inhibited mixtures. From these plots it is apparent that exposure to an inhibitor increased the reverse stutter variances above the 99th percentile line from the unchallenged plot for a majority (~56.67%) of treated mixtures, while leaving the allele variance and forward stutter variance largely unchanged or even lower than the regression line. A very small percentage (3.33%) of reverse stutter variance parameters exceeded the prior gamma 99th percentile line, and no allele or forward stutter variance data fell above this line.

Figure 4 shows the log_10_(variance parameter) v log_10_(Tc) plots for mixtures with an underestimated NOC, overlaid with the bands of expected variation from the unchallenged plots. Increased allele variance was observed for a substantial number of these mixtures; overall, ~28.30% of the mixtures had diagnostic values above the unchallenged 99th percentile line, mostly for higher level contributors, while the reverse and forward stutter variance plots tracked the unchallenged regression closely, with only ~3.77% of each data set above the unchallenged 99th percentile line. The only data above the line defined by the 99th percentile of the prior gamma distributions was a small percentage (~1.89%) of the allele variance diagnostic values.

Figure 5a–c are an overlay of the unchallenged bands of expected variation onto log_10_(variance parameter) v log_10_(Tc) plots for degraded mixtures. A moderate proportion of all three variance parameters for these mixtures (~21.11% for alleles, ~23.33% for reverse stutters, and ~12.22% for forward stutters) were above the unchallenged 99th percentile line. Much smaller proportions of each variance parameter exceeded the 99th percentile of the prior gamma distribution (6.67% for alleles, 2.22% for reverse stutters, and 1.11% for forward stutters).

Figure 6a–c are an overlay of the unchallenged bands of expected variation onto log_10_(variance parameter) v log_10_(Tc) plots for signal-saturated mixtures. These mixtures showed the most striking deviation from expectation, with ~40.00% of allele variance parameters, ~80.00% of reverse stutter variance parameters, and 70.00% of forward stutter diagnostics exceeding the unchallenged 99th percentile line. Many of these variances also exceeded the prior gamma distribution 99th percentile line (~45.00% for allele, ~75.00% for reverse stutter, and ~5.00% for forward stutter).

Figure 7a–c are an overlay of the unchallenged bands of expected variation onto modified variance diagnostic plots for mixtures of cell line DNA. As mentioned in the Methods section, these mixtures were excluded from the unchallenged data set due to unintuitive analysis outcomes that indicated STRmix was struggling to fit the observed data to the profile model. The plots indicated that allele modeling issues gave rise to the unsatisfactory analyses, demonstrating a population-wide shift upward in allele diagnostics while the reverse and forward stutter diagnostics remained largely unaffected. This was similar to the trend observed with underestimated NOC.

Table 11 summarizes the percentage of data points from each set that exceeded the unchallenged regression line, unchallenged 99th percentile line, prior gamma distribution mode line, and prior gamma distribution 99th percentile line. As a frame of reference, a data set closely tracking unchallenged expectations would have about half of its data points exceeding the line representing typical variances (regression line or prior gamma distribution mode line) and a small percentage of data points in excess of the line representing elevated variances (unchallenged 99th percentile line or prior gamma distribution 99th percentile line, respectively).

## 4. Discussion

The three STRmix variance parameters we have characterized in this paper fluctuate with increasing template amount, as well as with challenging amplification and/or interpretation conditions. The complexity of the log_10_(variance parameter) v log_10_(Tc) plots for the unchallenged data demonstrates the value of establishing an empirically determined working range for STRmix variance parameters instead of assuming that the observed average variance parameter values will always align with the prior gamma distribution.

While this work focused primarily on pragmatic applications of variance parameter data, it is also useful to theorize about what biological or profile modeling factors may have produced the observed patterns in the data. For instance, one might ask why the allele variance parameters for the unchallenged interpretations remained relatively centered on the prior gamma mode across the Tc range, while most of the unchallenged reverse and forward stutter variance parameters (95.28% for reverse stutter and 99.76% for forward stutter) were above the prior mode. These contrasting trends can be attributed to differences in peak detection for alleles, reverse stutters, and forward stutters. While peak height for all detected allelic data is considered during deconvolution, most reverse stutters at low Tc levels will be undetectable, requiring STRmix to model reverse stutter dropout and thus potentially causing the reverse stutter variance to land above the prior mode. The major inflection point of the reverse stutter plot, at a log_10_(Tc) value of ~2.79, or a Tc of ~617 RFU, corresponds to the approximate point at which stutters begin to be detected. Considering a typical range of reverse stutter ratios to be ~0.05 to 0.1, a Tc of ~617 RFU would equate to reverse stutter peak heights in the range of ~31–62 RFU, which straddles our detection thresholds of 35–71 RFU. Notice that beyond log_10_(Tc) values of ~2.79, the reverse stutter variance parameters once again move upward, while the allele variance parameters stay relatively flat. A plausible explanation for this trend is that expected stutter peak heights in STRmix are determined by static per-allele stutter ratios and therefore do not have the same degree of model flexibility as expected allele peak heights, which vary with the STRmix template parameter during interpretation. Similar to the reverse stutter variance at log_10_(Tc) value of ~2.79, the forward stutter variance parameter values begin significantly trending down at a log_10_(Tc) value of ~3, or a Tc value of ~1000 RFU, which would equate to forward stutter peak heights of ~1–50 RFU (typical forward stutter ratio range of ~0.001 to 0.05), again straddling our detection thresholds and suggesting a change in the modeling fit.

The log_10_(variance parameter) v log_10_(Tc) plots of the various challenged datasets also show distinct data patterns that can be associated with biological causes or elements of the STRmix profile model. For the inhibited data set, the most affected of the three variance parameters was reverse stutter, likely because the inhibition required STRmix to model undetected reverse stutter peaks across the profile due to reduced locus yields; in contrast, most of the alleles were still detected at the affected loci. More moderate but significant effects on the reverse stutter variance parameters were observed in the degraded data, which also requires STRmix to model undetected reverse stutter at higher molecular weight loci as peak heights decrease with degradation. However, moderate effects on the allele variance parameters were also observed; these effects are attributable to the difficulty of modeling high levels of degradation, particularly if the value of the exponential decay term in the STRmix degradation model approaches its user-defined ceiling (which occurred with many of the highly degraded mixtures in this set) [14]. The cell line and underestimated NOC data have similar effects on the variance parameters, in that the allele variance was the most affected of the three. This is a sensible result, given the apparent allele modeling issues with the cell line data and the intralocus allele imbalances that may result from NOC underestimation. Signal saturation, meanwhile, often had a pronounced effect on all three variance parameters. At the peak heights observed in saturated mixtures, tolerance for any peak height deviation from expectation is extremely low, and such deviation is more likely with the loss of linearity between peak height and template, leading to a cascade of effects on the variance parameters. However, not all of the saturated mixtures resulted in elevated variances, because there was variation both in the total number of off-scale peaks detected and the degree of saturation. The more nuanced variance expectations we have presented here can be useful in determining whether the extent of signal saturation observed in a profile has had a discernable effect on its interpretation.

As an example of how a working range for variance parameters might be implemented, Figure 8 shows the prior gamma distributions for allele, reverse stutter, and forward stutter variance parameters, overlaid with the corresponding parameters for a cell line mixture that resulted in an LR of 0 for the true minor contributor. While the electropherogram for this mixture was unremarkable (see Appendix A), the allele variance parameter was slightly elevated compared to the 99th percentile of our unchallenged log_10_(allele variance parameter) v log_10_(Tc) regression, which serves as a prompt for closer scrutiny of the interpretation, as well as contributing to an explanation for the aberrant LR result. Figure 9 and Figure 10 are two further examples of how the unchallenged regression data could be applied for routine use in the assessment of STRmix variance parameters from a case result. The interpretations assessed in both figures are from the inhibited mixture data set. In both cases, the allele variance parameter is below the 99th percentile of the unchallenged regression, but in Figure 9, neither stutter parameter is flagged as high, while in Figure 10 both are flagged. Notice in particular how similar the reverse stutter parameters are between the two interpretations; despite this similarity, a higher threshold for the reverse stutter variance parameter was applied to the 2-person data because it had a higher Tc than the 3-person data.

Despite the focus in this study on more precisely defining typical ranges for variance parameters, it is important to point out that the observation of a variance parameter outside of the norm does not by itself invalidate a STRmix interpretation; rather, it indicates that more variation than usual was needed for profile modeling. While the final LR is not necessarily a direct measure of how well an interpretation reflects the true contributors’ genotypes, it is notable that the variance outliers of the 3308 interpretations conducted in our study included only one instance of a false exclusion (i.e., an LR of 0 for a true contributor where Hp = true contributor + N-1 unrelated unknown contributors and Hd = N unrelated unknown contributors). The data giving rise to the false exclusion was for the 870 pg 9:1 mixture from the cell line data set featured in Figure 8. In this case, higher variances, more consistent with the trend in Figure 7, would have been necessary to capture the allele peak height variation observed in this mixture and avoid the false exclusion. This points to cell line DNA mixtures as potentially inappropriate validation samples for mixture data calibrated to casework-type samples.

While the variance parameter thresholds presented here easily lend themselves to the imposition of binary definitions of “good” and “bad” data, these labels are not appropriate to apply in such a rigid manner. As secondary diagnostics, variances are intended to encourage closer inspection of the input peak data as well as the results of the interpretation, i.e., in this instance, the genotype combinations that STRmix determined to be acceptable explanations of the electropherogram in question. However, despite the utility of secondary diagnostics as indicators of challenged input data, analyst appraisal of the electropherogram data and primary diagnostics can and should be the key measures by which interpretation reliability is assessed.

## 5. Conclusions

We have presented information relevant to the utilization and interpretation of three important secondary diagnostics contained in the output of the probabilistic genotyping software STRmix v2.8, specifically the allele variance parameter, reverse stutter variance parameter, and forward stutter variance parameter. Despite each of these being universal model parameters that are applied to data at all loci simultaneously, they behave differently with changes in DNA template, as well as challenged amplification and interpretation conditions. We found that the effect of increasing Tc on variance parameters, which was more readily visualized by taking the logarithm of both variables, was nonlinear and required fourth-order polynomial regression to achieve a satisfactory fit. These nonlinear regressions of variance parameters to Tc allowed us to semi-quantitatively compare the variance parameters of a STRmix interpretation to thresholds developed with unchallenged data in order to make more finely tuned appraisals of whether the average variance parameter values on a particular STRmix report are elevated. However, regardless of the benchmarks used to assess whether the variances for a particular interpretation are typical or elevated, as secondary diagnostics they are not intended to be hard analysis stop points. Instead, they are supplementary information to assist in examining the core of the STRmix output, which is the distribution of genotype combination weights.

## Figures and Tables

**Figure 1 genes-14-00102-f001:**
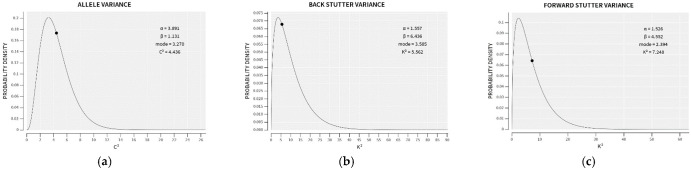
Examples of prior gamma distributions for the allele variance (**a**), reverse stutter variance (**b**), and forward stutter variance (**c**) parameters found in a STRmix v2.8 Interpretation Report. The average value of each variance parameter for the completed interpretation is indicated with a black dot on each distribution.

**Figure 2 genes-14-00102-f002:**
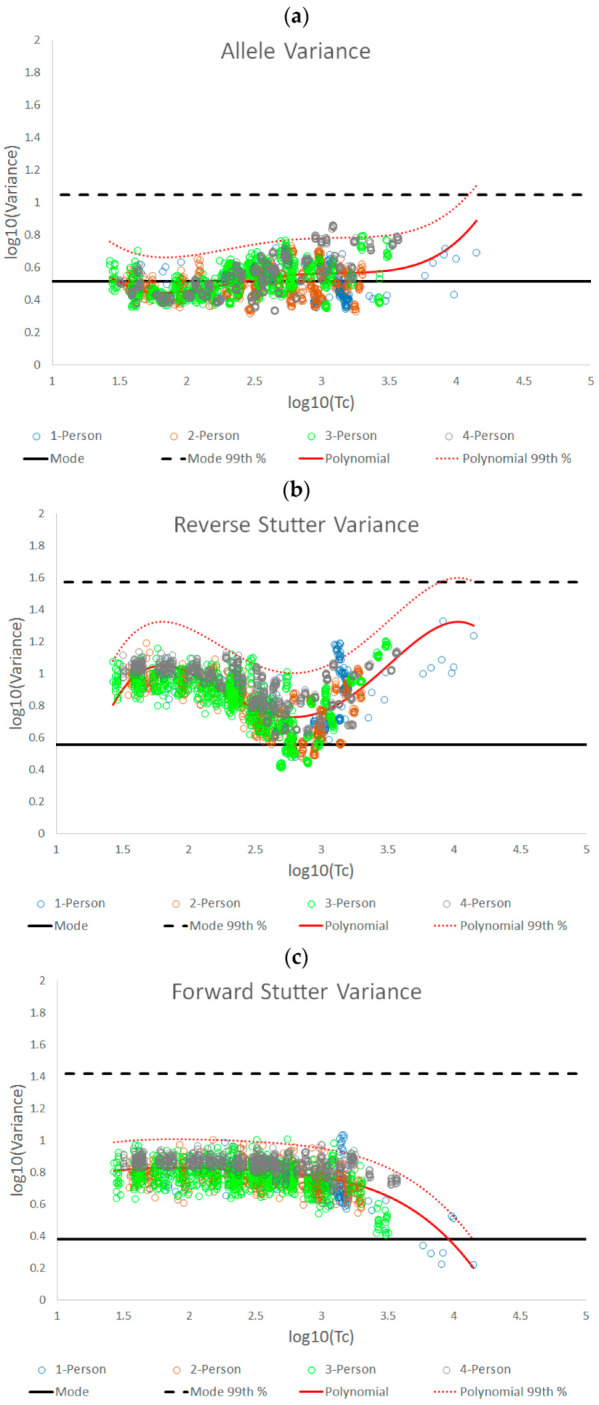
Plots of log_10_(variance parameter) v log_10_(Tc) for allele variance (**a**), reverse stutter variance (**b**), and forward stutter variance (**c**) of the unchallenged data set, overlaid with fourth-order polynomial regression lines (red solid lines) and the prior gamma modes (solid black lines). Additionally, shown are the 99th percentiles of the normal distributions around the fourth-order polynomial regressions (dotted red lines) and the 99th percentiles of the prior gamma distributions for the variance parameters (dotted black lines).

**Figure 3 genes-14-00102-f003:**
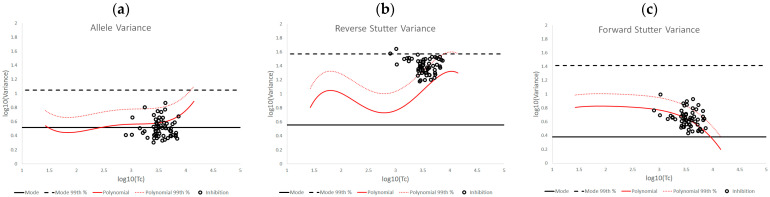
Plots of log_10_(variance parameter) v log_10_(Tc) for allele variance (**a**), reverse stutter variance (**b**), and forward stutter variance (**c**) of the inhibited data set.

**Figure 4 genes-14-00102-f004:**
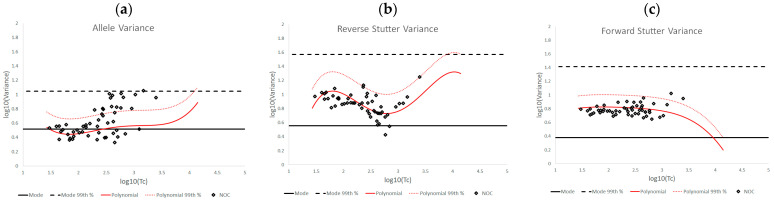
Plots of log_10_(variance parameter) v log_10_(Tc) for allele variance (**a**), reverse stutter variance (**b**), and forward stutter variance (**c**) of the underestimated NOC data set.

**Figure 5 genes-14-00102-f005:**
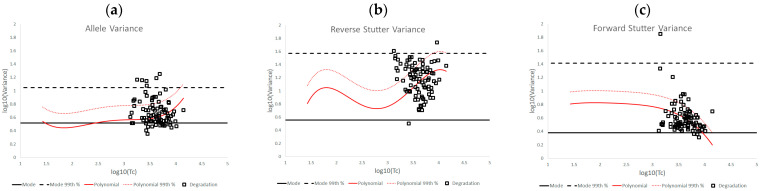
Plots of log_10_(variance parameter) v log_10_(Tc) for allele variance (**a**), reverse stutter variance (**b**), and forward stutter variance (**c**) of the degraded data set.

**Figure 6 genes-14-00102-f006:**
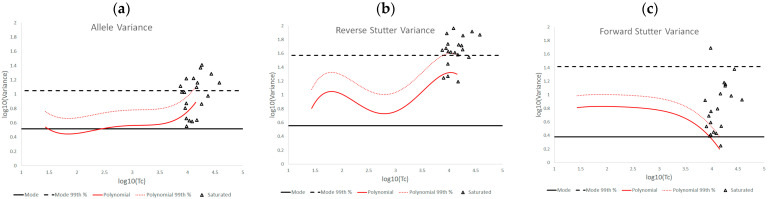
Plots of log_10_(variance parameter) v log_10_(Tc) for allele variance (**a**), reverse stutter variance (**b**), and forward stutter variance (**c**) of the signal-saturated data set.

**Figure 7 genes-14-00102-f007:**
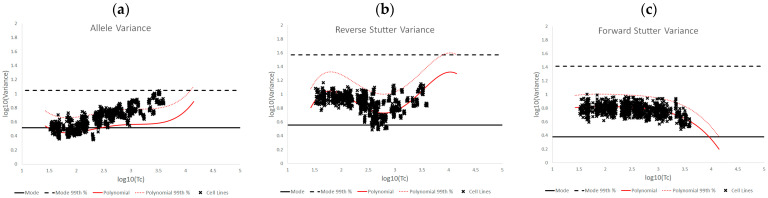
Plots of log_10_(variance parameter) v log_10_(Tc) for allele variance (**a**), reverse stutter variance (**b**), and forward stutter variance (**c**) of the cell line DNA mixture set.

**Figure 8 genes-14-00102-f008:**
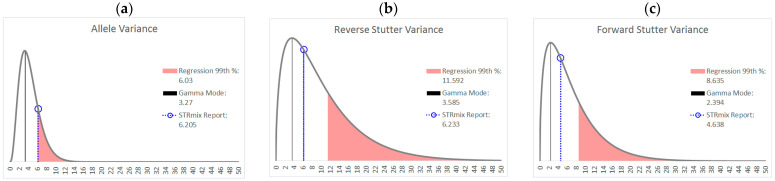
Prior gamma distributions for allele (**a**), reverse stutter (**b**) and forward stutter (**c**) variance parameters, along with the prior modes and the STRmix Interpretation Report variance parameter values for an 870 pg 9:1 mixture that resulted in a 0 LR for the true minor contributor. The allele variance parameter for the interpretation is flagged as high in this case.

**Figure 9 genes-14-00102-f009:**
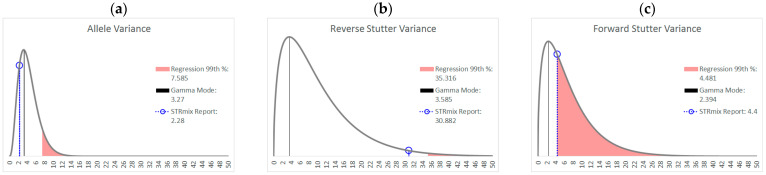
Prior gamma distributions for allele (**a**), reverse stutter (**b**) and forward stutter (**c**) variance parameters, along with the prior modes and the STRmix Interpretation Report variance parameter values for a 3 ng 2-person 10:1 mixture spiked with 475 µM hematin (Tc = 6882). None of the three variance parameters is flagged as high in this case.

**Figure 10 genes-14-00102-f010:**
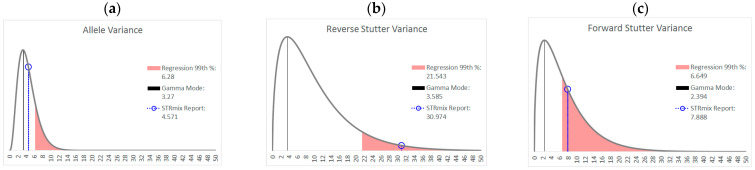
Prior gamma distributions for allele (**a**), reverse stutter (**b**) and forward stutter (**c**) variance parameters, along with the prior modes and the STRmix Interpretation Report variance parameter values for a 3 ng 4-person 10:5:2:1 mixture spiked with 625 µM hematin (Tc = 4097). Both the reverse and forward stutter variance parameters are flagged as high in this case.

**Table 11 genes-14-00102-t011:** Percentage of variance parameter values exceeding bands of expected variation for allele variance, reverse stutter variance, and forward stutter variance under unchallenged and challenged amplification and/or interpretation conditions. The first two benchmarks for each variance type are based on information in Table 9, while the third and fourth benchmarks are based on information in Table 10.

	% Greater Than…	Unchallenged	Inhibited	NOC −1 or −2	Degraded	Saturated	Cell Lines
Allele Variance	Polynomial Regression	48.90%	21.67%	67.92%	53.33%	55.00%	90.50%
	99th Percentile	1.05%	3.33%	28.30%	21.11%	40.00%	21.84%
	Mode	47.09%	41.67%	58.49%	84.44%	100.00%	80.38%
	99th Percentile	0.00%	0.00%	1.89%	6.67%	45.00%	0.00%
Reverse Stutter Variance	Polynomial Regression	50.91%	100.00%	39.62%	57.78%	85.00%	45.60%
	99th Percentile	1.15%	56.67%	3.77%	23.33%	80.00%	1.92%
	Mode	95.28%	100.00%	96.23%	98.89%	100.00%	97.37%
	99th Percentile	0.00%	3.33%	0.00%	2.22%	75.00%	0.00%
Forward Stutter Variance	Polynomial Regression	53.15%	48.33%	28.30%	44.44%	100.00%	37.71%
	99th Percentile	1.00%	16.67%	3.77%	12.22%	70.00%	0.51%
	Mode	99.76%	100.00%	100.00%	96.67%	95.00%	100.00%
	99th Percentile	0.00%	0.00%	0.00%	1.11%	5.00%	0.00%

## Data Availability

The data presented in this study are available in the Appendix A listed above.

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
