# Peer review of "Improving the Utilization of STRmix™ Variance Parameters as Semi-Quantitative Profile Modeling Metrics"

_genes, 2022, doi:10.3390/genes14010102_

Round 1

Reviewer 1 Report

The authors have written on the use of the variance parameters in STRmix as a diagnostic to aid in the evaluation of a deconvolution. The paper is well written and uses appropriate language. The breadth of work they have carried out is impressive and there is some very useful information in their findings. There are however some issues with the way the data has been displayed and interpreted, which I feel need to be addressed. I detail these below. I think that once these issues have been resolved that the publication will be of general interest to the forensic community.

General point 1:

The way in which the data is transformed and graphed gives the impression of a linear relationship that may not be present. The issue comes from the fact that one variable has been transformed using another, and then the two are plotted against each other. You can show the problem with doing this yourself using excel; in column A fill 1000 cells with random numbers between 1 and 2. In column B fill 1000 cells with random numbers between 1000 and 10 000. In column C divide the values in column A by those in column B.  This is now a setup that is very similar to that the authors have where column A is the variance parameter, column B is Tc and column C is the transformed variance (i.e., var/Tc). If you plot column B against column C it will appear as though there is a trend between the two (when both axes are on a log10 scale this will appear linear). Note that this has occurred even though the numbers generated were completely random and had no correlation with each other.

For the paper, what this means is that the regressions and diagonal plots in Figs 2 to 7 could be misleading given the way they are calculated and displayed. In this instance I believe the better way to display the data would be just considering the distribution of values for the unadjusted variance parameter. This could be plotted over the prior distribution of variance values (see something akin to this in Fig 4 of “Factors affecting peak height variability for short tandem repeat data” by Taylor et al 2016, but perhaps both as line graphs). Each of the lines on the current plots still have their equivalents in the distribution style plot:

-          Prior distribution mode - black solid line on current plot – vertical line at prior mode in distribution plot

-          Prior distribution 99th percentile - black dashed line on current plot – vertical line at prior 99th percentile in distribution plot

-          Regression  - red solid line on current plot – vertical line at mean of observed variance in distribution plot

-          3SD of residuals  - red dashed line on current plot – vertical line at 99.7th percentile of observed variance in distribution plot  (99.7% would be the equivalent of 3SD but you may wish to just plot the 99th percentile to be consistent with the values displayed for the prior)

In some instances, you have shown that there is an effect of peak height on the variance (beyond the trend that is present simply due to the way the variables are being plotted), for example in Fig 2b.  The distribution plots I suggest above would not show this information (as all height data is combined in the distribution. The appropriate way to display this would be (in addition to the distribution plot) to plot the unadjusted variance parameter value (y-axis) against Tc (x-axis). The mode and 99th percentile of the prior distribution will be shown as horizontal lines. Then a regression can be done on this dataset (i.e., var parameter vs Tc) and the regression line & points shown on the plot. In the instance of Fig 2b you will then see a trend in the observed var values across Tc that has some non-zero slope. You can even then use the standard error on the slope term to draw a conclusion as to whether the trend is significantly different from 0 (i.e., whether Tc is really playing a role).

General point 2:

In the discussion it would be good to have a section that talks about the reasons the authors hypothesise for the observations. For example:

-          saturated profiles showing high variance values – due to saturated peaks no longer following the linear trend between template and fluorescence.

-          there is a trend in variance over template in Fig 2b – could it be due to lower template samples having to account for stutter dropout more that higher ones (rather than this actually being caused by some other imbalance factor unique to stutters)?

Forming these sorts of hypotheses can then lead to further investigation e.g., to investigate the latter example one could plot the variance against the number of stutter peaks that had dropped out.

Minor points:

When displaying formulae with ‘log’ terms I find it useful to include the base i.e., ‘log10’ as many people will automatically think of base-e.

There is various STRmix documentation that refers to the variance constant. This comes about from earlier versions of STRmix when the value was indeed a constant throughout the analysis (and in fact across all analyses). At some point this constant was changed to be a parameter in the model i.e., its value changes during the analysis, and between analyses. It is a bit funny to talk about a variable constant, and so a better term to use is the variance parameter.

Author Response

Please see the attached Word document for our response to your review.

Reviewer 2 Report

Improving the utilization of STRmix™ variance constants as 2 semi-quantitative profile modeling metrics

Kyle Duke, Steven Myers, Daniela Cuenca, and Jeanette Wallin

This paper examines a function of the variance constants output from a commonly used PG software STRmix™.  They find a reasonably strong inference:

1.      From the back stutter variance for the inhibited data,

2.      From the allele data for underestimated NoC,

3.      From back stutter and allele for the degraded data,

4.      From back, forward and allele for saturated data

5.      From back, forward and allele for cell line data

The authors speculate on the use of thresholds as a tool for diagnostic but one could also envisage a bayesian inference approach.

Ln 125 See Supplemental Table 1 for anonymized contributor genotypes.

Not for the paper but by way of conversation:  Is this a substitution code?  For example is the letter A at a locus always the same allele for the different donors?  I suspect so.  Is A always a smaller allele than B?  If this is so the code is probably breakable in very short order.  I personally am concerned you are close to disclosing genotypes.  I do not know the legality of that in California.  Maybe in the paper point out that even with these coded genotypes you cannot work out things like overlap of allele and stutter. 

ln 146 to one or two contributors less than the true

I don’t care whether the authors change this or not as the meaning is clear but grammatically less should be fewer.

Ln 148 contributor number due to allele overlap between contributors and/or allelic dropout

This intrigues me.  It may imply that this was done with knowledge of the references.  I do think I can sometimes infer this from the mixed profile alone but certainly not always hence I suspect use of the references.  I think that is OK in this study but it is highly inappropriate in casework.  May I ask the authors to please be a little clearer here what was done and if that is use of the references state a caution against doing that in casework.

Figure 2 (and the others).  The y axis label is log(variance/Tc).  Tc is defined in the text as “DNA amount” but I think it is actually the strmix t variable for the largest contributor.  This is the template estimate (rfu) at molecular weight x where x is the offset.  The variance, I think, is actually the variance constant c2.  This is termed “modified variance diagnostic” which is probably better than the actual y axis label.  May I ask the authors to consider changing.

Table 11.  May I be helped to understand please.  Why are the modified variance diagnostics for forward and back stutter to often above the mode?  Is this a function of dividing by Tc? 

Ln 393 not to the need for a hard cap on variance constant values.

I don’t understand this clause.  May I be helped or could the authors help in the text?

Author Response

(The authors gave the same response as above.)
